# The Causal Relationship between the Morning Chronotype and the Gut Microbiota: A Bidirectional Two-Sample Mendelian Randomization Study

**DOI:** 10.3390/nu16010046

**Published:** 2023-12-22

**Authors:** Manman Chen, Zhenghe Wang, Din Son Tan, Xijie Wang, Zichen Ye, Zhilan Xie, Daqian Zhang, Dandan Wu, Yuankai Zhao, Yimin Qu, Yu Jiang

**Affiliations:** 1School of Population Medicine and Public Health, Chinese Academy of Medical Sciences and Peking Union Medical College, Beijing 100730, China; chenmm@pumc.edu.cn (M.C.); zichenye@sph.pumc.edu.cn (Z.Y.); xzl_719@163.com (Z.X.); zhangdaqian@sph.pumc.edu.cn (D.Z.); wudandan@sph.pumc.edu.cn (D.W.); z4311652@163.com (Y.Z.); quyimin@sph.pumc.edu.cn (Y.Q.); 2Department of Epidemiology, School of Public Health, Southern Medical University, Guangzhou 510515, China; wzh116086@smu.edu.cn; 3Vanke School of Public Health and Institute for Healthy China, Tsinghua University, Beijing 100084, China; hongchen_huajun@163.com

**Keywords:** morning chronotype, gut microbiota, bidirectional Mendelian randomization, single-nucleotide polymorphisms

## Abstract

Background: Numerous observational studies have documented an association between the circadian rhythm and the composition of the gut microbiota. However, the bidirectional causal effect of the morning chronotype on the gut microbiota is unknown. Methods: A two-sample Mendelian randomization study was performed, using the summary statistics of the morning chronotype from the European Consortium and those of the gut microbiota from the largest available genome-wide association study meta-analysis, conducted by the MiBioGen consortium. The inverse variance-weighted (IVW), weighted mode, weighted median, MR-Egger regression, and simple mode methods were used to examine the causal association between the morning chronotype and the gut microbiota. A reverse Mendelian randomization analysis was conducted on the gut microbiota, which was identified as causally linked to the morning chronotype in the initial Mendelian randomization analysis. Cochran’s Q statistics were employed to assess the heterogeneity of the instrumental variables. Results: Inverse variance-weighted estimates suggested that the morning chronotype had a protective effect on Family *Bacteroidaceae* (*β* = −0.072; 95% CI: −0.143, −0.001; *p* = 0.047), Genus *Parabacteroides* (*β* = −0.112; 95% CI: −0.184, −0.039; *p* = 0.002), and Genus *Bacteroides* (*β* = −0.072; 95% CI: −0.143, −0.001; *p* = 0.047). In addition, the gut microbiota (Family *Bacteroidaceae* (OR = 0.925; 95% CI: 0.857, 0.999; *p* = 0.047), Genus *Parabacteroides* (OR = 0.915; 95% CI: 0.858, 0.975; *p* = 0.007), and Genus *Bacteroides* (OR = 0.925; 95% CI: 0.857, 0.999; *p* = 0.047)) demonstrated positive effects on the morning chronotype. No significant heterogeneity in the instrumental variables, or in horizontal pleiotropy, was found. Conclusion: This two-sample Mendelian randomization study found that Family *Bacteroidaceae*, Genus *Parabacteroides*, and Genus *Bacteroides* were causally associated with the morning chronotype. Further randomized controlled trials are needed to clarify the effects of the gut microbiota on the morning chronotype, as well as their specific protective mechanisms.

## 1. Introduction

The morning chronotype, characterized by a natural inclination to wake up and be active in the early hours of the day, is a fundamental aspect of human chronobiology [1]. It represents an individual’s preference for being more active and alert during specific times of the day, with morning chronotype individuals favoring the early hours [2]. This preference is governed by the circadian clock, a complex system that regulates the body’s internal processes over a 24 h cycle [3,4]. Morning chronotype individuals often experience a natural synchrony with the external environment, as their peak performance aligns with typical daytime activities [5,6]. Their unique chronotype predisposes them to various circadian-related traits and behaviors that can profoundly affect their overall health and well-being [7,8]. There is compelling evidence suggesting a connection between disruptions in the circadian rhythm and the onset of diseases, with a notable focus on metabolic and psychiatric disorders [9,10]. Recent research underscores the intricate and compelling connection between an individual’s mood and their chronotype, suggesting that our daily rhythms not only influence our sleep–wake patterns, but also play pivotal roles in shaping our emotional well-being and mental states [11,12,13].

The human gut microbiome is a crucial factor in both the maintenance of health and the emergence of diseases in humans [14]. At the same time, the gut microbiota, comprising trillions of microorganisms residing in the digestive system, has been recognized for its profound implications in various aspects of human health [15,16,17]. Recent studies have linked alterations in the composition of the gut microbiota with the onset and progression of several diseases [18,19]. Circadian rhythms manifest in nearly every organism, governing various aspects of biological and physiological processes. There exists a bidirectional circadian interaction between the host and its gut microbiota, and potential circadian orchestration of both the host and its gut microbiota in response to invading pathogens [20]. Disruptions to the circadian system can lead to alterations in the composition of microbiome communities, consequently impacting the host’s metabolism, energy regulation, and inflammatory pathways, and thus play a role in the development of metabolic syndrome. Previous research has shown that two SCFA-produced genera, *Lachnospiraceae UCG004 and Odoribacter,* promote extended sleep duration, while the order *Selenomonadales* and the class *Negativicutes* are associated with insomnia risk [21].

Mendelian randomization (MR) leverages genetic variants to create instrumental variables for assessing the causal link between exposure to and the outcome of a disease [22]. MR has been extensively utilized to investigate the causal links between the gut microbiota and various diseases, encompassing both metabolic and autoimmune conditions [23,24]. An association between the gut microbiota and the sleep phenotype was found in a recent Mendelian randomization study, but no inverse relationship was demonstrated [21]. Thus, understanding the relationship between the morning chronotype and the gut microbiota represents a burgeoning field of research that holds the significant potential to advance our knowledge of human biology and health. Our primary objective was to initiate an MR study through which to confirm the causal influence of the gut microbiota taxa on the morning chronotype, and examined the potential bidirectional nature of this causal effect.

## 2. Materials and Methods

We obtained de-identified summary-level data from publicly accessible GWAS studies. The dataset focusing on the morning chronotype was derived from a collaboration within the European Consortium [25], which ensured a comprehensive representation of this particular trait. Simultaneously, the dataset encompassing information about the gut microbiota was obtained from a GWAS study conducted under the aegis of the international MiBioGen consortium initiative [26,27], reflecting a global perspective on this intricate field of research. It is important to emphasize that ethical and regulatory guidelines were rigorously upheld throughout this data collection endeavor. Each cohort contributing to the GWAS studies diligently obtained ethical approval and consent from their respective participants, reinforcing the ethical integrity of the research process. Our two-sample MR investigation was conducted following the framework illustrated in Figure 1.

### 2.1. The Morning Chronotype

To facilitate our analysis of genome-wide significant variants, we implemented a binary phenotype utilizing the same data field as that employed in the chronotype study. In this classification, we categorized participants based on their self-reported chronotype preferences [25]. Those who expressed a clear inclination towards being “Definitely an ‘evening’ person” or identified as “More of an ‘evening’ than ‘morning’ person” were grouped as controls. On the contrary, individuals who firmly characterized themselves as “Definitely a ‘morning’ person” or leaned towards “More of a ‘morning’ than ‘evening’ person” were classified as cases. Participants who provided responses such as “Do not know” or “Prefer not to answer” were considered as missing data, ensuring a comprehensive and robust analysis. This binary classification strategy encompassed a substantial cohort of 403,195 participants participating in the GWAS. Within this cohort, there were 252,287 cases and 150,908 controls, contributing to the depth and reliability of our investigation.

### 2.2. Gut Microbiota

Genetic instruments for gut microbiota were derived from a comprehensive association study encompassing 24 cohorts, consisting of 18,340 participants [26]. The studies with these cohorts were conducted in various countries, including the USA, Canada, Israel, Republic of Korea, Germany, Denmark, The Netherlands, Belgium, Sweden, Finland, and the UK. Out of the 24 cohorts, the majority focused on individuals of single ancestry, primarily those of European descent (16 cohorts, *n* = 13,266). Notably, 17 of the 24 cohorts (*n* = 13,804) had participants with mean ages ranging from 50 to 62 years. In the quantitative microbiome trait loci (mbQTL) mapping analysis for each cohort, only taxa that appeared in more than 10% of the samples were considered, resulting in a dataset comprising 211 taxa (131 genera, 35 families, 20 orders, 16 classes, and 9 phyla). Meanwhile, the binary trait loci mapping (mbBTL) analysis included taxa that appeared in a percentage range of 10% to 90% in the included samples. It is worth noting that all of the included cohorts incorporated adjustments for covariates related to sex and age in their calculations. The summary-level statistics from this association study are publicly accessible on the www.mibiogen.org website.

### 2.3. Genetic Instrument Selection

Single-nucleotide polymorphisms (SNPs) associated with the morning chronotype at the genome-wide significance level (*p*  <  5  ×  10^−8^) were extracted from the European ancestries of 403,195 individuals (Figure 2).

In order to ensure the credibility and precision of our conclusions regarding the causal connection between the gut microbiome and the morning chronotype, a series of rigorous quality control measures were implemented to curate the most suitable instrumental variables (IVs) [28]. Initially, SNPs that exhibited significant relationships with the gut microbiome were chosen as IVs. Two distinct thresholds were applied in this selection process. The first threshold involved the identification of SNPs with statistical significance levels below the genome-wide threshold (*p*  <  5  ×  10^−8^) to serve as IVs. Unfortunately, this initial threshold yielded only a limited number of gut microbiota-related IVs. To explore a more comprehensive spectrum of potential causal associations between the morning chronotype and the gut microbiota, a second threshold was introduced. This threshold involved the selection of SNPs with significance levels smaller than the locus-wide threshold (*p* < 1.0 × 10^–5^) as the second set of IVs [23]. Following this, we utilized the PLINK method to perform SNP clumping, following stringent linkage disequilibrium (LD) criteria, with an R^2^ < 0.001 threshold and genomic windows exceeding 10,000 kb, to guarantee the independence of our genetic instrumental variables. For situations in which SNPs displayed LD, we retained the one with the most significant *p*-value.

### 2.4. Statistical Analysis

Five popular MR methods were used for features containing multiple IVs: inverse variance-weighted (IVW) test [29], weighted mode [30], weighted median [31], the MR-Egger regression [32], and simple mode methods [33]. The IVW method is noted to exhibit slightly greater power than the others in specific circumstances [31]. As a consequence, for results involving multiple instrumental variables, the primary approach relied on the IVW method, while the other four methods were employed to provide supplementary insights.

To assess the robustness of the results, several sensitivity analyses were performed. A leave-one-out analysis was conducted to ascertain whether a single SNP was responsible for driving the causal signal. This approach involves a comparison of the variance explained by the instrumental variables for both the exposure and the outcome. We conducted a heterogeneity test using Cochran’s Q statistics and the two-sample MR package across the instrumental variables. Significant Q statistics with a *p*-value < 0.05 may suggest the existence of heterogeneity [34,35]. Additionally, we computed *F* statistics to assess the presence of weak instrument bias [36]. The strength of each IV was assessed by calculating its *F*-statistic using the formula $F=\frac{R^{2}(N-2)}{1-R^{2}}$, where *R*^2^ represents the proportion of variance in the exposure explained by the genetic variants, and *N* represents sample size [37]. Any *F*-value below 10 was considered indicative of a weak instrument, which was subsequently excluded from the analysis.

In our quest to investigate the potential causal influence of the gut microbiota on the identified morning chronotype, we carried out a reverse MR analysis. In this reverse MR analysis, we considered the gut microbiota as the exposure and the identified morning chronotype as the outcome, utilizing SNPs associated with gut microbiota as IVs. To assess the directionality of causality, we employed the MR Steiger directionality test [38]. The MR analyses were performed in the R version 4.0.2 computing environment, using the latest TwoSampleMR (https://github.com/MRCIEU/TwoSampleMR, accessed on date (20 September 2023)) packages.

## 3. Results

### 3.1. Genetic Instruments of Exposure and Outcome

Following the initial SNP selection, in which a stringent significance threshold of *p* < 5 × 10^–8^ was applied, we performed pairwise LD clumping and aligned coding alleles between the exposure and outcome summary statistics. This process enabled us to identify instrumental variables (IVs) that met the three core Mendelian randomization (MR) assumptions. In our study on the morning chronotype, we analyzed data from 403,195 participants, among whom 62.6% (252,287 individuals) self-identified as morning people. Ultimately, 122 SNPs were employed as IVs for assessing the morning chronotype, and find detailed information about these IVs can be found in Appendix A. In a similar vein, for our investigation on the gut microbiota, we conducted SNP selection with a significance threshold of *p* < 1 × 10^–5^, followed by pairwise LD clumping and coding allele alignment, to identify valid IVs satisfying the three fundamental MR assumptions. This phase of the study comprised 18,340 individuals drawn from 24 different cohorts, with the majority (13,266) having European ancestry. In accordance with our IV selection criteria, we utilized a total of 235 SNPs as instrumental variables for the analysis of the gut microbiota, and you can refer to Appendix A for comprehensive details on the selected instrumental variables.

### 3.2. MR of the Morning Chronotype and the Gut Microbiota

The results, as displayed in Figure 3, indicate a positive association between the morning chronotype and several microbial taxonomic categories—Phylum *Lentisphaerae* (*β* = 0.146; 95% CI: 0.007, 0.285; *p* = 0.039); Class *Bacilli* (*β* = 0.082; 95% CI: 0.010, 0.154; *p* = 0.025); Class *Lentisphaeria* (*β* = 0.151; 95% CI: 0.012, 0.290; *p* = 0.033); Order *Lactobacillales* (*β* = 0.085; 95% CI: 0.013, 0.157; *p* = 0.021); Order *Victivallales* (*β* = 0.151; 95% CI: 0.012, 0.290; *p* = 0.033); Family *Prevotellaceae* (*β* = 0.092; 95% CI: 0.014, 0.17; *p* = 0.020); Family *Streptococcaceae* (*β* = 0.077; 95% CI: 0.004, 0.151; *p* = 0.039); Genus *Eggerthella* (*β* = 0.171; 95% CI: 0.028, 0.315; *p* = 0.019); and Genus *Streptococcus* (*β* = 0.076; 95% CI: 0.002, 0.151; *p* = 0.043)—using the IVW method. Conversely, a negative association was observed between the morning chronotype and the following microbial taxonomic categories using the IVW method: Class *Deltaproteobacteria* (*β* = −0.080; 95% CI: −0.157, −0.003; *p* = 0.043); Order *Desulfovibrionales* (*β* = −0.143; 95% CI: −0.277, −0.008; *p* = 0.038); Family *Desulfovibrionaceae* (*β* = −0.079; 95% CI: −0.157, −0.002; *p* = 0.045); Family *Porphyromonadaceae* (*β* = −0.08; 95% CI: −0.157, −0.003; *p* = 0.043); Family *Prevotellaceae* (*β* = −0.086; 95% CI: −0.158, −0.014; *p* = 0.019); Family *Bacteroidaceae* (*β* = −0.072; 95% CI: −0.143, −0.001; *p* = 0.047); Family *Rikenellaceae* (*β* = −0.091; 95% CI: −0.162, −0.019; *p* = 0.013); Genus *Parabacteroides* (*β* = −0.112; 95% CI: −0.184, −0.039; *p* = 0.002); Genus *Alistipes* (*β* = −0.090; 95% CI: −0.169, −0.010; *p* = 0.026); and Genus *Bacteroides* (*β* = −0.072; 95% CI: −0.143, −0.001; *p* = 0.047).

The weighted median, MR-Egger regression, weighted mode and simple mode methods that provided evidence of the relationship between the morning chronotype and the gut microbiota are shown in Appendix A. In addition, Figure 4, Figure 5 and Figure 6 show scatter plots of the SNP-outcome associations against the SNP-exposure associations. Appendix A showcase Mendelian randomization plots that elucidated the connection between the gut microbiota and the morning chronotype. Our rigorous analysis, employing advanced statistical methods, provides a robust confirmation of the accuracy and directionality of the causal effects. The MR Steiger directionality test, as presented in Appendix A, affirms that the causal effects align with our expectations, bolstering the validity of our findings. Furthermore, our investigation into horizontal pleiotropy, conducted through MR-Egger regression analyses, yielded no substantial evidence of its presence across all analyses, as demonstrated in Appendix A. Additionally, our study identified evidence for heterogeneity among the causal effects, as indicated with Cochran’s Q statistic. This statistical evidence, with a significance level of *p* for Cochran’s Q < 0.05, implies that there are variations in the estimated causal effects. This finding, detailed in Appendix A, adds depth to our understanding of the complexity and diversity of the causal relationships under investigation. In summary, our comprehensive analysis employs a range of methods to ensure the reliability and validity of our results, providing a solid foundation for the conclusions drawn from our study.

### 3.3. Reverse MR of the Gut Microbiota and the Morning Chronotype

According to the results of reverse MR analysis, there was a suggestive association between the gut microbiota and the morning chronotype (Figure 7). Specifically, the IVW method provided evidence that certain gut microbiota—Class *Erysipelotrichia* (OR: 1.074; 95% CI: 1.018, 1.134; *p* = 0.009); Class *Negativicutes* (OR: 1.073; 95% CI: 1.003, 1.47; *p* = 0.039); Order *Enterobacteriales* (OR: 1.090; 95% CI: 1.026, 1.157; *p* = 0.005); Order *Erysipelotrichales* (OR: 1.074; 95% CI: 1.018, 1.134; *p* = 0.009); Order *Selenomonadales* (OR: 1.073; 95% CI: 1.003, 1.147; *p* = 0.039); Family *Enterobacteriaceae* (OR: 1.090; 95% CI: 1.026, 1.157; *p* = 0.005); Family *Erysipelotrichaceae* (OR: 1.074; 95% CI: 1.018, 1.134; *p* = 0.009); Family *Peptococcaceae* (OR: 1.049; 95% CI: 1.008, 1.091; *p* = 0.019); Genus *Catenibacterium* (OR: 1.040; 95% CI: 1.005, 1.075; *p* = 0.025); Genus *Intestinibacter* (OR: 1.044; 95% CI: 1.006, 1.084; *p* = 0.024); Genus *Tyzzerella3* (OR: 1.050; 95% CI: 1.017, 1.084; *p* = 0.003); and Genus *Victivallis* (OR: 1.028; 95% CI: 1.002, 1.054; *p* = 0.032)—were positively associated with the morning chronotype. However, Family *Bacteroidaceae* (OR: 0.925; 95% CI: 0.857, 0.999; *p* = 0.047); Genus *Alloprevotella* (OR: 0.966; 95% CI: 0.936, 0.998; *p* = 0.037); Genus *Bacteroides* (OR: 0.925; 95% CI: 0.857, 0.999; *p* = 0.047); Genus *Bifidobacterium* (OR: 0.951; 95% CI: 0.910, 0.994; *p* = 0.026); Genus *Parabacteroides* (OR: 0.915; 95% CI: 0.858, 0.975; *p* = 0.007); Genus *Prevotella7* (OR: 0.972; 95% CI: 0.949, 0.996; *p* = 0.020); and Genus *Ruminococcus1* (OR: 0.942; 95% CI: 0.896, 0.990; *p* = 0.019) were found to be negatively associated with the morning chronotype.

Evidence of a connection between the gut microbiota and the morning chronotype was demonstrated through the weighted median, MR-Egger regression, weighted mode, and simple mode methods in this study (Appendix A). In addition, Figure 8, Figure 9 and Figure 10 show scatter plots of the SNP-outcome associations against the SNP-exposure associations. Appendix A showcase Mendelian randomization plots that elucidated the connection between the gut microbiota and the morning chronotype. The MR Steiger directionality test validated the correctness of the causal effect directions (Appendix A). Across all analyses in the MR-Egger regression, there was no substantial evidence of horizontal pleiotropy (Appendix A). Additionally, we did observe evidence of heterogeneity among the causal effects, as indicated with Cochran’s Q statistic (i.e., *p* for Cochran’s Q < 0.05) in the IVW model (Appendix A). To address potential outlier SNPs, we conducted the MR-PRESSO test and found no outlier SNPs.

## 4. Discussion

In this study, we employed summary statistics of the morning chronotype and summary statistics of the gut microbiota, sourced from the MiBioGen consortium’s largest GWAS meta-analysis. Our approach involved conducting a two-sample MR analysis to investigate the potential causal relationship between the morning chronotype and the gut microbiota. Our analysis revealed causal associations between the morning chronotype and certain microbial taxa, including those of Family *Bacteroidaceae*, Genus *Parabacteroides*, and Genus *Bacteroides*. To gain a more comprehensive understanding of the effects of the morning chronotype on the gut microbiota and its underlying mechanisms, additional randomized controlled trials (RCTs) are warranted.

In recent years, there has been a growing focus on the potential contributions of the gut microbiota to human well-being [39]. In our study, we found that Family *Bacteroidaceae*, Genus *Parabacteroides*, and Genus *Bacteroides* were casually associated with the morning chronotype. Importantly, individuals with a morning chronotype have the capacity to decrease the population of their gut microbiota. There are studies consistent with the results of this study [40,41]. Some studies have found that a strong correlation was observed between *Bacteroidaceae* and *hematopoietic* damage markers, suggesting that an increased abundance of *Bacteroidaceae* could potentially contribute to the *hematopoietic* toxicity induced by benzene [42]. In contrast, *Bacteroidaceae* establish multi-organ connections between sites of intestinal inflammation and distal bone marrow, promoting the proliferation and differentiation of myeloid cells specialized in intestinal tissue repair [41].

Within the intricate community of the gut microbiota, particular attention has been directed toward the bacterial genus *Parabacteroides*. Our study suggests that being a morning person may be associated with a reduction in the abundance of *Parabacteroides*. However, a previous study found that the abundance of *Parabacteroides* was inversely associated with obesity, especially in the female and middle-aged populations [43]. A possible reason for this is that *Parabacteroides* modulate the host metabolism by increasing the production of the secondary bile acids and succinate [44]. There is currently no clear explanation for the association between *Parabacteroides* and being a morning person. Further research is needed to unravel the intricate interactions and underlying mechanisms that might elucidate this intriguing relationship and shed light on the biological and genetic factors influencing one’s chronotype.

The maintenance of intestinal barrier function depends on the balance of pathogenic bacteria and probiotics [45]. In recent research, it has come to light that various *Bacteroides* species play pivotal roles in promoting gut homeostasis through the secretion of immunomodulatory factors [46,47]. These beneficial microorganisms have been found to actively contribute to the delicate balance within the gastrointestinal system by releasing compounds that help regulate the immune response and maintain a harmonious environment in the gut [48]. Nevertheless, it is important to note that certain *Bacteroides* species can exhibit a dual nature, with both beneficial and potentially harmful roles, depending on their specific locations within the host [49]. Current studies on the human gut microbiome have indicated the significant involvement of microbiota in the initiation of different forms of cancer in humans, and recent research has suggested that microbiota play pivotal roles in the genesis of various types of cancer [50,51,52]. Our research has revealed a compelling link between being a morning person and a decrease in Bacteroides abundance, and future investigations may offer a more nuanced understanding of these connections and their potential implications for health. Potential factors include the fact that a microbiome imbalance can lead to a greater susceptibility to cancer, as pathogens are capable of exerting detrimental effects on the host’s physiology, metabolism, and immune system, consequently promoting the growth of tumors [53].

This study boasts several strengths, including the utilization of MR analysis to establish the causal link between the gut microbiota and the morning chronotype. Genetic variants associated with gut microbiota were derived from the most extensive GWAS meta-analysis available, ensuring the robustness of the instruments in the MR analysis. To address potential issues, like horizontal pleiotropy, the study employed tests such as MR-PRESSO and the MR-Egger regression intercept term. However, it is crucial to acknowledge certain limitations while interpreting the results. The reliance on summary statistics instead of raw data precluded the possibility of conducting subgroup analyses, marking a constraint in the study’s scope. Since the lowest taxonomic level in the exposure dataset was genus, this restriction prevented us from further exploring the causal association between the gut microbiota and the morning chronotype at the species level. However, this taxonomic restriction underscores the need for more detailed information at the species level to better understand the nuanced relationship between the composition of the gut microbiota and the expression of morning chronotype traits. Subsequent studies should contemplate employing metagenomic sequencing techniques on cohorts with distinctly assigned chronotypes to investigate the correlation between the morning chronotype and the gut microbiota [54].

## 5. Conclusions

In summary, our comprehensive analysis identified a causal association between being a morning person and having specific microbial taxa in the gut, such as Family *Bacteroidaceae*, Genus *Parabacteroides*, and Genus *Bacteroides*. This noteworthy discovery underscores the intricate interplay between the morning chronotype and the composition of the gut microbiota, serving as a poignant reminder that our health is a complex tapestry, intricately woven from the diverse threads of lifestyle, genetics, and environmental factors. Furthermore, the association we studied should also increase the necessity for rigorous randomized controlled trials through which to better understand the impact of the morning chronotype on the gut microbiota and its mechanisms. 

## Figures and Tables

**Figure 1 nutrients-16-00046-f001:**
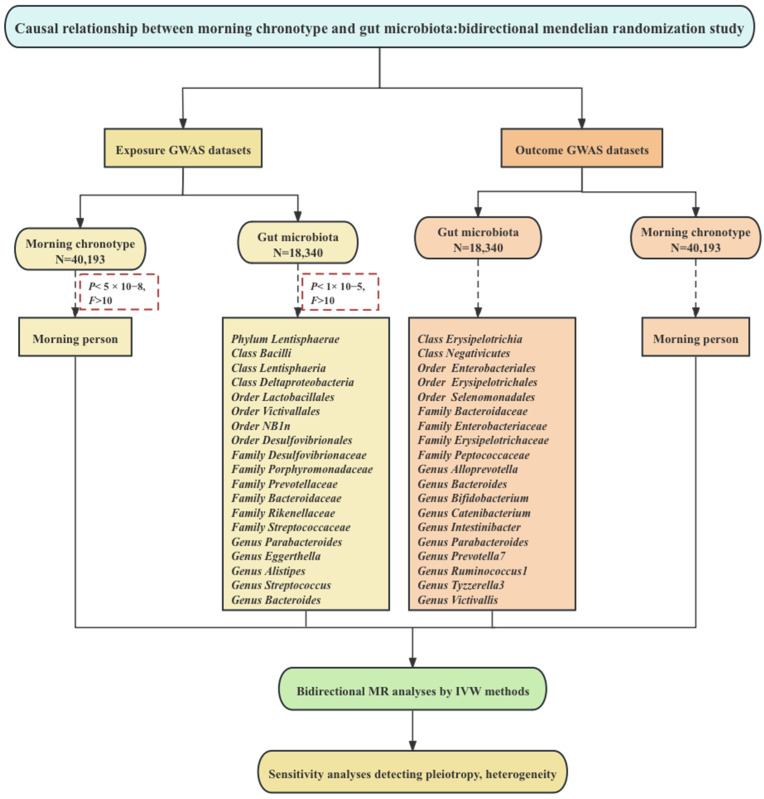
Study design and framework. Note: Mendelian randomization necessitates the use of valid genetic instrumental variables that adhere to three essential assumptions. These include Assumption 1, which implies that genetic variants should predict exposure; Assumption 2, requiring that genetic variants remain independent of confounding factors; and Assumption 3, which asserts that genetic variants influence the outcome exclusively through exposure, rather than through alternative pathways.

**Figure 2 nutrients-16-00046-f002:**
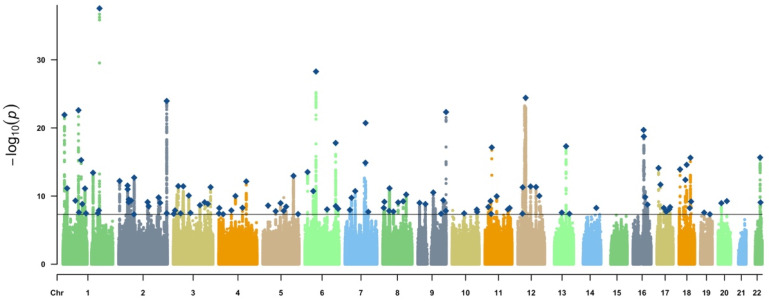
Manhattan plot depicting the morning chronotype. Note: chr, chromosome; The solid gray line indicates the typical genome-wide significance threshold of *p* = 5 × 10^−8^ identified through permutation testing. Lead variants are annotated with a diamond.

**Figure 3 nutrients-16-00046-f003:**
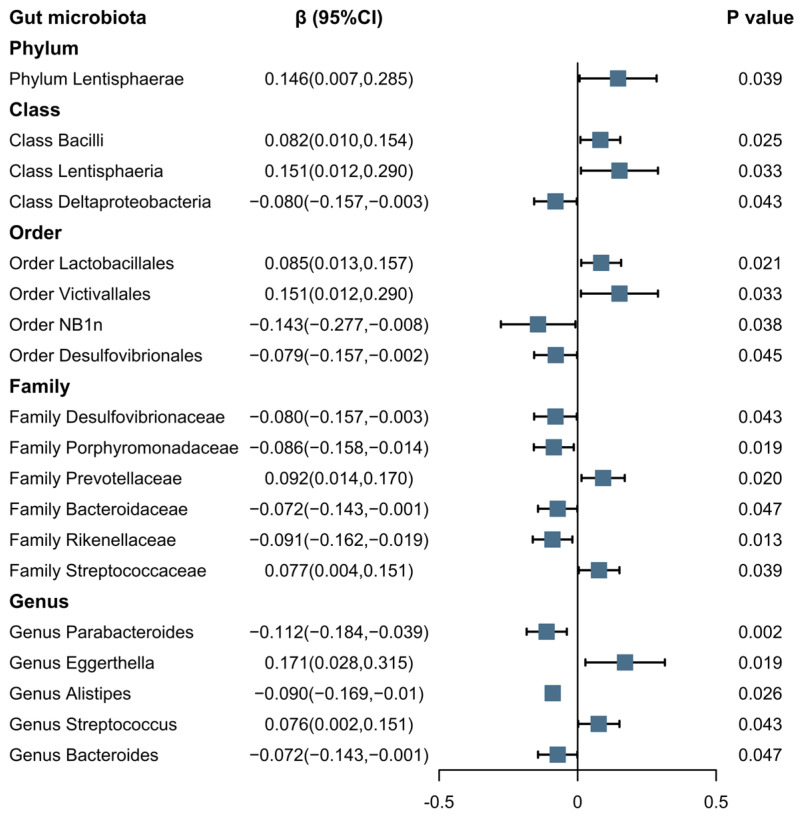
The association between the morning chronotype and the gut microbiota using IVW methods. Note: Blue solid squares represent OR values.

**Figure 4 nutrients-16-00046-f004:**
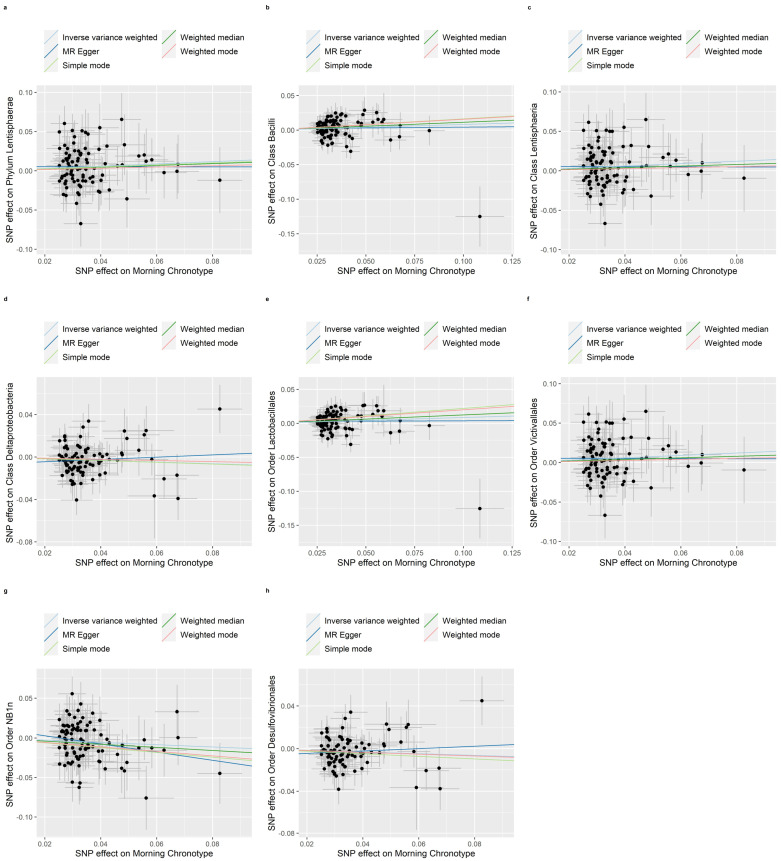
Scatter plots in which the SNP-gut microbiota (phyla, classes, and orders) associations are plotted against the SNP-morning chronotype associations. (**a**) Phylum Lentisphaerae; (**b**) Class Bacilli; (**c**) Class Lentisphaeria; (**d**) Class Deltaproteobacteria; (**e**) Order Lactobacillales; (**f**) Order Victivallales; (**g**) Order NB1n; (**h**) Order Desulfovibrionales. Note: black dots represent effects; grey lines represent confidence interval.

**Figure 5 nutrients-16-00046-f005:**
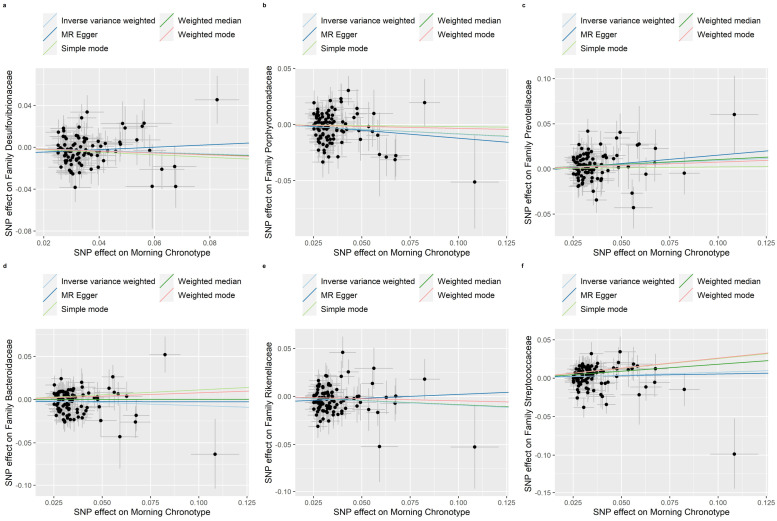
Scatter plots in which the SNP-gut microbiota (families) associations are plotted against the SNP-morning chronotype associations. (**a**) Family Desulfovibrionaceae; (**b**) Family Porphyromonadaceae; (**c**) Family Prevotellaceae; (**d**) Family Bacteroidaceae; (**e**) Family Rikenellaceae; (**f**) Family Streptococcaceae. Note: black dots represent effects; grey lines represent confidence interval.

**Figure 6 nutrients-16-00046-f006:**
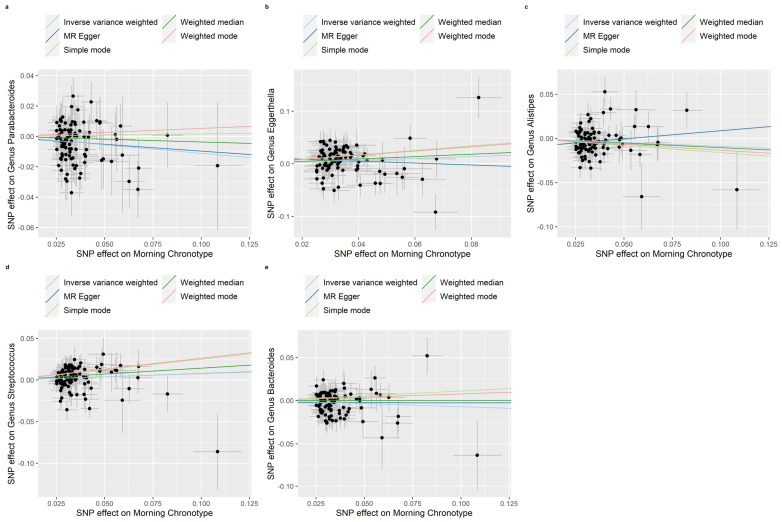
Scatter plots in which the SNP-gut microbiota (genera) associations are plotted against the SNP-morning chronotype associations. (**a**) Genus Parabacteroides; (**b**) Genus Eggerthella; (**c**) Genus Alistipes; (**d**) Genus Streptococcus; (**e**) Genus Bacteroides. Note: black dots represent effects; grey lines represent confidence interval.

**Figure 7 nutrients-16-00046-f007:**
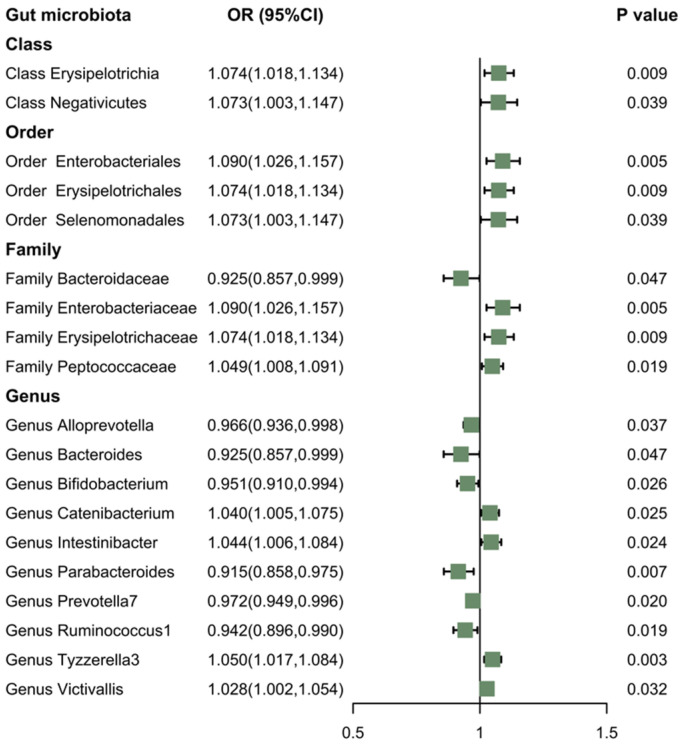
The association between the gut microbiota and the morning chronotype using IVW methods. Note: Green solid squares represent OR values.

**Figure 8 nutrients-16-00046-f008:**
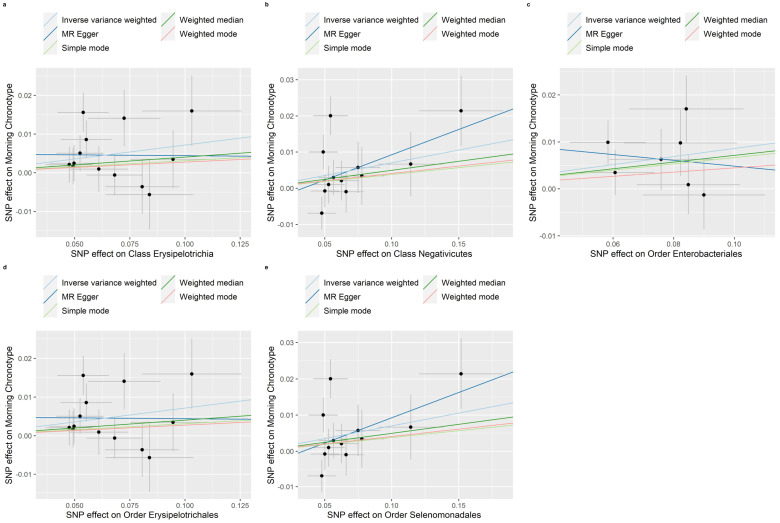
Scatter plots in which the SNP-morning chronotype associations are plotted against the SNP-gut microbiota (classes and orders) associations. (**a**) Class Erysipelotrichia; (**b**) Class Negativicutes; (**c**) Order Enterobacteriales; (**d**) Order Erysipelotrichales; (**e**) Order Selenomonadales. Note: black dots represent effects; grey lines represent confidence interval.

**Figure 9 nutrients-16-00046-f009:**
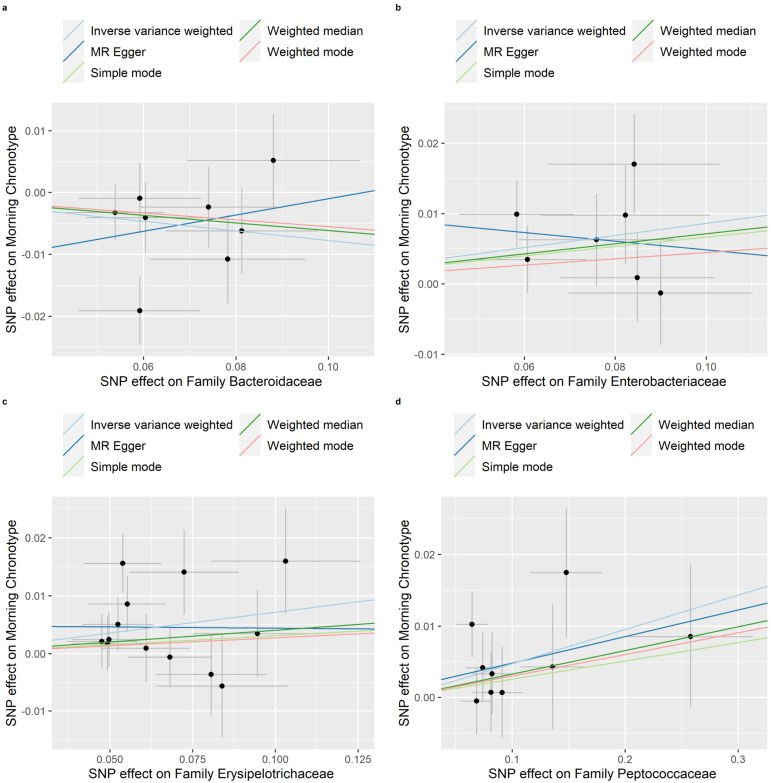
Scatter plots in which the SNP-morning chronotype associations are plotted against the SNP-gut microbiota (families) associations. (**a**) Family Bacteroidaceae; (**b**) Family Enterobacteriaceae; (**c**) Family Erysipelotrichaceae; (**d**) Family Peptococcaceae. Note: black dots represent effects; grey lines represent confidence interval.

**Figure 10 nutrients-16-00046-f010:**
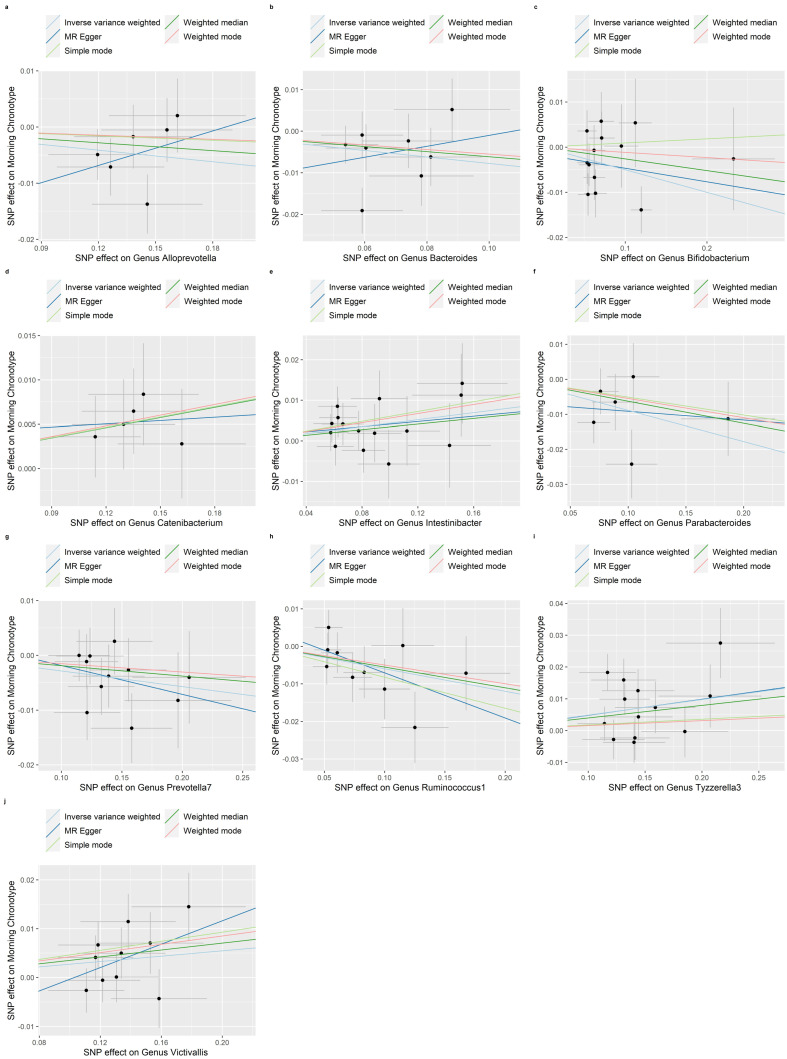
Scatter plots in which the SNP-morning chronotype associations are plotted against the SNP-gut microbiota (genera) associations. (**a**) Genus Alloprevotella; (**b**) Genus Bacteroides; (**c**) Genus Bifidobacterium; (**d**) Genus Catenibacterium; (**e**) Genus Intestinibacter; (**f**) Genus Parabacteroides; (**g**) Genus Prevotella7; (**h**) Genus Ruminococcus1; (**i**) Genus Tyzzerella3; (**j**) Genus Victivallis. Note: black dots represent effects; grey lines represent confidence interval.

## Data Availability

The datasets analyzed during the current study are available from the European Consortium [25], and the MiBioGen repository, https://mibiogen.gcc.rug.nl/ (accessed on date (25 September 2023)) [26].

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
