# Peer review of "The Causal Relationship between the Morning Chronotype and the Gut Microbiota: A Bidirectional Two-Sample Mendelian Randomization Study"

_nutrients, 2023, doi:10.3390/nu16010046_

Round 1
Reviewer 1 Report
Comments and Suggestions for Authors
The objective of Chen et al.'s research is to validate the causal relationship between taxa of intestinal microbiota and morning chronotype, as well as to investigate the possibility of a bidirectional effect via Mendelian randomization.
The findings were clearly shown, and the manuscript exhibited good writing and structure.
However, as the authors pointed out, this research has a significant limitation: the exposure dataset only included taxonomic levels reduced to the genus level.
Consequently, a comprehensive examination of the causal relationship between intestinal microbiota and morning chronotype at the species level is not possible.
Point 1. The research examines the possible bidirectional influence via Mendelian randomization in addition to examining the causal association between gut microbiota taxa and morning chronotype.
Points 2 and 3. The subject, in my opinion, is only moderately relevant. Given that one of the defining characteristics of the chronotype is its impact on sleep patterns, it bears some resemblance to the study "Causal Effects of Gut Microbiota on Sleep-Related Phenotypes: A Two-Sample Mendelian Randomization Study" (DOI: 10.3390/clockssleep5030037) by Yue et al. published in 2023. Additionally, the Authors should take into account the study, regardless of its metagenomics focus, titled "Metagenomic analysis reveals the distinctive characteristics of gut microbiota linked to human chronotypes" (https://doi.org/10.1096/fj.202100857RR) and integrate it into the discussion
Point 4. The limitation of the study is that the authors did not identify a causal association between specific species of gut microbiota and morning chronotype. The authors have not provided details regarding their intended approach to address this constraint in their study.
Point 5. The conclusions are coherent, although they only partially address the primary question.
Point 6. the references are appropriate.
Point 7. Even if there are high-resolution figures in the supplementary materials, the quality of those in the text should be improved.
Author Response
The objective of Chen et al.'s research is to validate the causal relationship between taxa of intestinal microbiota and morning chronotype, as well as to investigate the possibility of a bidirectional effect via Mendelian randomization.
The findings were clearly shown, and the manuscript exhibited good writing and structure.
However, as the authors pointed out, this research has a significant limitation: the exposure dataset only included taxonomic levels reduced to the genus level.
Consequently, a comprehensive examination of the causal relationship between intestinal microbiota and morning chronotype at the species level is not possible.
Response: Thank you for your thoughtful review of our study. We appreciate your acknowledgment of the clarity in presenting our findings and the overall quality of the manuscript's writing and structure. We also acknowledge the limitation you highlighted regarding the taxonomic levels in our exposure dataset, which was limited to the genus level.
Point 1. The research examines the possible bidirectional influence via Mendelian randomization in addition to examining the causal association between gut microbiota taxa and morning chronotype.
Response: Thank you for your valuable feedback. We appreciate your recognition of our efforts to explore the bidirectional association via Mendelian randomization in addition to investigating the causal association between gut microbiota taxa and morning chronotype. This aspect of the research aims to provide a more comprehensive understanding of the relationship between these variables.
Points 2 and 3. The subject, in my opinion, is only moderately relevant. Given that one of the defining characteristics of the chronotype is its impact on sleep patterns, it bears some resemblance to the study "Causal Effects of Gut Microbiota on Sleep-Related Phenotypes: A Two-Sample Mendelian Randomization Study" (DOI: 10.3390/clockssleep5030037) by Yue et al. published in 2023. Additionally, the Authors should take into account the study, regardless of its metagenomics focus, titled "Metagenomic analysis reveals the distinctive characteristics of gut microbiota linked to human chronotypes" (https://doi.org/10.1096/fj.202100857RR) and integrate it into the discussion.
Response: Thanks for your constructive suggestion. We appreciate your thoughtful feedback and have carefully considered your suggestions. We acknowledge the importance of addressing the impact of chronotype on sleep patterns, and we agree that it aligns with the broader context of research in this field. We have taken your comments to heart and made the necessary adjustments. As per your recommendation, we have included the specific references in the discussion. The specific additions are as follows:
Introduction: An association between gut microbiota and sleep phenotype was found in a recent Mendelian randomization study but no inverse relationship was demonstrated [21].
Discussion: Subsequent studies should contemplate employing metagenomic sequencing techniques on cohorts with distinctly assigned chronotypes to investigate the correlation between morning chronotype and gut microbiota [54].
Point 4. The limitation of the study is that the authors did not identify a causal association between specific species of gut microbiota and morning chronotype. The authors have not provided details regarding their intended approach to address this constraint in their study.
Response: Thank you for your insightful comments regarding the limitation in our study concerning the identification of a causal association between specific species of gut microbiota and morning chronotype. We obtained deidentified summary-level data from publicly accessible GWAS studies, and the gut microbiota dataset only comprising 211 taxa (131 genera, 35 families, 20 orders, 16 classes, and 9 phyla). Based on your comments it has been stated in the limitations. The revisions are as follows:
Since the lowest taxonomic level in the exposure dataset was genus, this restriction prevented us from further exploring the causal association between gut microbiota and morning chronotype at the species level. However, this taxonomic restriction underscores the need for more detailed information at the species level to better understand the nuanced relationship between gut microbiota composition and the expression of morning chronotype traits.
Point 5. The conclusions are coherent, although they only partially address the primary question.
Response: Thank you for your feedback on the coherence of our conclusions. We appreciate your acknowledgment of the clarity in our study's key takeaways.
Point 6. the references are appropriate.
Response: Thank you for confirming that the references in our manuscript are deemed appropriate. We strive to ensure the relevance and reliability of our cited sources.
Point 7. Even if there are high-resolution figures in the supplementary materials, the quality of those in the text should be improved.
Response: Thank you for highlighting the issue regarding the quality of figures in the text. We apologize for any inconvenience, and we appreciate your constructive feedback. In response to your suggestion, we will ensure to enhance the resolution and quality of the figures presented in the main text and supplementary materials. We have provided clearer visuals to improve the overall readability and interpretation of our results in the revised manuscript.
Reviewer 2 Report
Comments and Suggestions for Authors
Reviewer’s Comments
“Causal relationship between morning chronotype and gut microbiota: a bidirectional two-sample mendelian randomization study” presents latest information on the association between microbiota and chronotype. They demonstrated that Parabacteroide and Bacteroides were causally associated with morning chronotype.
Minor Modifications
1. References are required in the statement “The human gut microbiota is a crucial factor in both the maintenance of health and the emergence of diseases in human”.
2. There is a need for spacing between references and the ending word in a sentence. E.g. as in line 63 “health[14-16]” should be written as “health [14-16]”. Please revise throughout the entire work.
3. Fig. 4. “Scatter plots in which the SNP-gut microbiota associations are plotted against the SNP-morning chronotype associations” should be broken into 3 figures to enhance clarity of the figures.
4. Fig. 6. “Scatter plots in which the SNP-gut microbiota associations are plotted against the SNP-morning chronotype associations” should be broken into 3 figures to enhance clarity of the figures.
5. Line 295. “There are studies consistent with the results of this study.” Please cite those studies which are consistent with your findings as you have stated here.
6. Line 303. “This study conducted that morning person may reduce the abundance of Parabacteroides.” This sentence requires revision to ensure language agreement. Secondly, there is a need to cite reference to support the argument.
7. Lines 315-318. “These beneficial microorganisms have been found to actively contribute to the delicate balance within the gastrointestinal system by releasing compounds that help regulate the immune response and maintain a harmonious environment in the gut.” Please cite those studies from which this statement is derived.
8. Lines 320-324. Revise the following sentence into a single sentence since they both indicate the same idea. “Current studies on the human gut microbiome have indicated the significant involvement of microbiota in the initiation of different forms of cancer in humans. Recent research on the human gut microbiome has suggested that the microbiota play pivotal roles in the genesis of various types of cancer in humans[47-49].”
9. The conclusion “which subsequently lowers the risk of cancer” in line 325 cannot be made in this work. Please revise.
10. Please cite references to support your statement in lines 325-328.
11. The conclusion, although succinct, I recommend its expansion to include the utility of your findings to health and wellness in general.
Author Response
- References are required in the statement “The human gut microbiota is a crucial factor in both the maintenance of health and the emergence of diseases in human”.
Response: Thank you for your valuable feedback. We have carefully checked this sentence and revised the sentence. The specific revisions are as follows:
The human gut microbiota is a crucial factor in both the maintenance of health and the emergence of diseases in human [14].
- There is a need for spacing between references and the ending word in a sentence. E.g. as in line 63 “health[14-16]” should be written as “health [14-16]”. Please revise throughout the entire work.
Response: Thank you for your constructive feedback. We appreciate your attention to detail. We will promptly make the necessary revisions to include spacing between references and the ending word in sentences throughout the entire manuscript. This adjustment will enhance the overall readability and formatting of our work. The study has undergone revisions to address these issues.
- Fig. 4. “Scatter plots in which the SNP-gut microbiota associations are plotted against the SNP-morning chronotype associations” should be broken into 3 figures to enhance clarity of the figures.
Response: Thank you for your valuable suggestion. We appreciate your input and recognize the importance of enhancing the clarity of Fig. 4. Following your recommendation, we have divided Fig. 4 into three separate figures to provide a more focused and clearer presentation of the SNP-gut microbiota associations. This modification will contribute to a better understanding of the relationships illustrated in the scatter plots. The specific revisions are as follows:
Fig. 4. Scatter plots in which the SNP-gut microbiota (phyla, classes, and orders) associations are plotted against the SNP-morning chronotype associations
Fig. 5 Scatter plots in which the SNP-gut microbiota (families) associations are plotted against the SNP-morning chronotype associations
Fig. 6 Scatter plots in which the SNP-gut microbiota (genera) associations are plotted against the SNP-morning chronotype associations
- Fig. 6. “Scatter plots in which the SNP-gut microbiota associations are plotted against the SNP-morning chronotype associations” should be broken into 3 figures to enhance clarity of the figures.
Response: Thank you for your valuable suggestion. We appreciate your input and recognize the importance of enhancing the clarity of Fig. 6. Following your recommendation, we have divided Fig. 6 into three separate figures to provide a more focused and clearer presentation of the SNP-gut microbiota associations. This modification will contribute to a better understanding of the relationships illustrated in the scatter plots. The specific revisions are as follows:
Fig. 8. Scatter plots in which the SNP-morning chronotype associations are plotted against the SNP-gut microbiota (classes and orders) associations
Fig. 9. Scatter plots in which the SNP-morning chronotype associations are plotted against the SNP-gut microbiota (families) associations
Fig. 10. Scatter plots in which the SNP-morning chronotype associations are plotted against the SNP-gut microbiota (genera) associations
- Line 295. “There are studies consistent with the results of this study.” Please cite those studies which are consistent with your findings as you have stated here.
Response: Thank you for your valuable feedback. We have carefully checked this sentence and added appropriate references. The specific revisions are as follows:
There are studies consistent with the results of this study [40-41].
- Line 303. “This study conducted that morning person may reduce the abundance of Parabacteroides.” This sentence requires revision to ensure language agreement. Secondly, there is a need to cite reference to support the argument.
Response: Thank you for your valuable comments. We will revise the sentence for language agreement and include a reference to support the argument. The revised sentence could read:
Our study suggests that being a morning person may be associated with a reduction in the abundance of Parabacteroides.
- Lines 315-318. “These beneficial microorganisms have been found to actively contribute to the delicate balance within the gastrointestinal system by releasing compounds that help regulate the immune response and maintain a harmonious environment in the gut.” Please cite those studies from which this statement is derived.
Response: Thank you for your feedback. We acknowledge the importance of providing proper references to support our statements. The revised sentence with appropriate citation could be:
These beneficial microorganisms have been found to actively contribute to the delicate balance within the gastrointestinal system by releasing compounds that help regulate the immune response and maintain a harmonious environment in the gut [48].
- Lines 320-324. Revise the following sentence into a single sentence since they both indicate the same idea. “Current studies on the human gut microbiome have indicated the significant involvement of microbiota in the initiation of different forms of cancer in humans. Recent research on the human gut microbiome has suggested that the microbiota play pivotal roles in the genesis of various types of cancer in humans[47-49].”
Response: Thanks for your constructive suggestion. We appreciate the suggestion to combine the sentences for conciseness.
Current studies on the human gut microbiome have indicated the significant involvement of microbiota in the initiation of different forms of cancer in humans, and recent research has suggested that the microbiota play pivotal roles in the genesis of various types of cancer [50-52].
- The conclusion “which subsequently lowers the risk of cancer” in line 325 cannot be made in this work. Please revise.
Response: Thanks for your constructive suggestion. We acknowledge the importance of accurately reflecting the study's findings in the conclusion. The revisions are as follows:
Our research has revealed a compelling link between being a morning person and a decrease in Bacteroides abundance, and future investigations may offer a more nuanced understanding of these connections and their potential implications for health.
- Please cite references to support your statement in lines 325-328.
Response: Thank you for your valuable comments. The specific revisions are as follows:
The potential factors were that microbiome imbalance can lead to a greater susceptibility to cancer, as pathogens are capable of exerting detrimental effects on the host's physiology, metabolism, and immune system, consequently promoting the growth of tumors [53].
- The conclusion, although succinct, I recommend its expansion to include the utility of your findings to health and wellness in general.
Response: Thank you for your valuable feedback. We appreciate your suggestion to expand the conclusion to encompass the broader utility of our findings in the context of health and wellness. The specific revisions are as follows:
In summary, our comprehensive analysis identified causal association between being a morning person and specific microbial taxa in the gut, such as Family Bacteroidaceae, Genus Parabacteroides, and Genus Bacteroides. This noteworthy discovery under-scores the intricate interplay between the morning chronotype and the gut microbiota composition, serving as a poignant reminder that our health is a complex tapestry intricately woven from diverse threads of lifestyle, genetics, and environmental factors. Furthermore, the association we studied should also increase the necessity for rigorous randomized controlled trials to better understand the impact of morning chronotype on gut microbiota and its mechanisms.
Round 2
Reviewer 1 Report
Comments and Suggestions for Authors
The revised manuscript is now suitable for publication.
Author Response
We sincerely appreciate your careful review of our manuscript and are delighted to receive your positive feedback.